# Segment, Select, Correct: A Framework for Weakly-Supervised Referring Segmentation

## Abstract

Referring Image Segmentation (RIS) – the problem of identifying objects in images through natural language sentences – is a challenging task currently mostly solved through supervised learning. However, while collecting referred annotation masks is a time-consuming process, the few existing weakly-supervised and zero-shot approaches fall significantly short in performance compared to fully-supervised learning ones. To bridge the performance gap without mask annotations, we propose a novel weakly-supervised framework that tackles RIS by decomposing it into three steps: obtaining instance masks for the object mentioned in the referencing instruction (*segment*), using zero-shot learning to select a potentially correct mask for the given instruction (*select*), and bootstrapping a model which allows for fixing the mistakes of zero-shot selection (*correct*). In our experiments, using only the first two steps (zero-shot segment and select) outperforms other zero-shot baselines by as much as $19\%$, while our full method improves upon this much stronger baseline and sets the new state-of-the-art for weakly-supervised RIS, reducing the gap between the weakly-supervised and fully-supervised methods in some cases from around 33% to as little as 14%.

## 1 Introduction

Identifying particular object instances in images using natural language expressions – defined in the literature as referring image segmentation (RIS) (Wang et al., 2022b; Yang et al., 2022; Wu et al., 2022; Yu et al., 2023) – is an important problem that has many real-world applications including autonomous driving, general human-robot interactions (Wang et al., 2019) or natural language-driven image editing (Chen et al., 2018) to name a few. This problem is typically solved by training large vision and language models using supervised data from datasets of `image`, `referring expressions` and `referred mask` triplets (Hu et al., 2016; Yang et al., 2022).

However, collecting the required annotation masks for this task can be difficult, since annotating dense prediction masks given referring expressions is a time consuming process. Existing weakly-supervised (Strudel et al., 2022) and zero-shot (Yu et al., 2023) approaches attempt to address this problem by eliminating the need for using these masks, yet their performance is significantly worse than fully-supervised learning alternatives.

In this work, we tackle the problem of learning a weakly-supervised referring image segmentation model by leveraging the insight that fundamentally the problem can be divided into two steps: (i) obtaining instance masks for the desired object class referred in the expression (e.g., given the sentence *"the car on the left of the person"* the desired object class is **car**), and (ii) choosing the right mask from the ones obtained based on the referencing instruction (e.g., in the previous example it should be the **car** that is *"on the left of the person"* instead of any other ones in the image).

To solve (i), we design an open-vocabulary instance segmentation for referring expressions that generates all instance segmentation masks for that object. Given an accurate selection mechanism, we could solve (ii) directly, and to achieve this we first propose a zero-shot step based on work by Yang et al. (2023). However, this mechanism – as the CLIP-based zero-shot selection proposed by Yu et al. (2023) – makes mistakes which significantly reduce the performance of the overall system, despite the fact that (i) generates strong candidate masks. To tackle this problem, we propose a corrective step that trains a model to perform weakly-supervised RIS. This step pre-trains a model using the

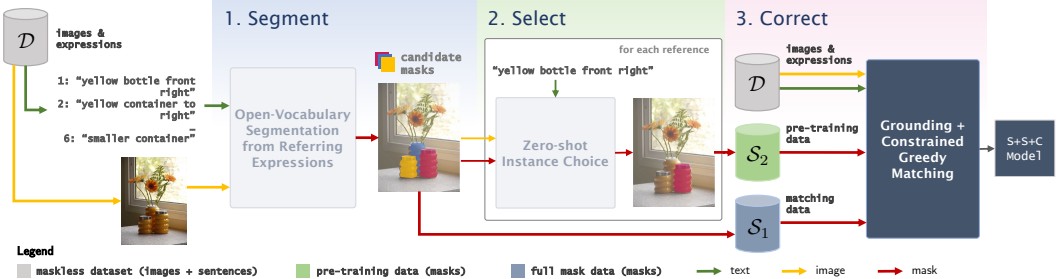

Figure 1: ***Segment, Select, Correct* for Referring Image Segmentation**: our three stage approach consists of using an open-vocabulary segmentation step from referring expressions to obtain all the candidate masks for the object in those sentences (*segment*, Stage 1), followed by a zero-shot instance choice module to select the most likely right mask (*select*, Stage 2), and then training a corrected RIS model using constrained greedy matching to fix the zero-shot mistakes (*correct*, Stage 3).

zero-shot selected masks from step (ii) and corrects potential mistakes using a constrained greedy matching scheme. Our full method is summarized in Figure 1.

Our main contributions are: (1) we introduce *segment*, *select*, *correct* (S+S+C) as a three-stage framework to perform referring image segmentation **without supervised referring masks** by training a model on pseudo-masks obtained using a zero-shot pipeline; (2) we establish new state-of-the-art performance in both zero-shot and weakly-supervised RIS, outperforming the zero-shot method by Yu et al. (2023) by as much as $19\%$, and the weakly-supervised methods by Liu et al. (2023a); Kim et al. (2023) by significant margins (up to $26\%$) in most testing sets in RefCOCO (Yu et al., 2016), RefCOCO+ (Mao et al., 2016) and RefCOCOg (Nagaraja et al., 2016). Finally, we highlight the benefits of our design choices in a series of ablations of the stages of the framework.

## 2 PRELIMINARIES AND RELATED WORK

### 2.1 PROBLEM SETUP AND NOTATION

Formally, the objective of referring image segmentation is to obtain a model $f : \mathbb{R}^{\mathcal{I}} \times \mathcal{T} \to [0, 1]^{\mathcal{I}}$, where for a given input image $\mathbf{I} \in \mathbb{R}^{\mathcal{I}}$ and an expression $\mathbf{T} \in \mathcal{T}$ the model outputs a binary, pixel-level mask of $1$ where the referred object in $\mathbf{T}$ exists, and $0$ elsewhere. Most of the existing literature treat this as a supervised learning problem, taking a dataset of image, referring sentences and segmentation mask pairs, i.e., $(\mathbf{I}_i, \mathbf{T}_i, \mathbf{M}_i)$, and training a text-conditioned segmentation pipeline end-to-end using a binary cross-entropy loss (Wang et al., 2022b; Wu et al., 2022; Yang et al., 2022). Training and testing datasets commonly used include RefCOCO (Yu et al., 2016), RefCOCO+ (Mao et al., 2016), and RefCOCOg (Nagaraja et al., 2016).

Crucially to our work, these datasets contain implicitly more information that is not leveraged in any previous work which comes from the dataset building process. For example, on building RefCOCO, the authors from Yu et al. (2016) started with an image from the COCO dataset (Lin et al., 2014), selected $2 - 3$ segmentation masks from objects in that image, and asked users to create 3 sentences referring to that specific instance of the object in the frame. In practice, this means that for each image $\mathbf{I}_i$ in the dataset we have a set $\left\{ \mathbf{M}_{i,j}, \{\mathbf{T}_{i,j,k}\}_{k=1}^{n_{i,j}^{\mathbf{T}}} \right\}$ where for each object mask, $\mathbf{M}_{i,j}$, for all $k$, $\mathbf{T}_{i,j,k}$ are references to the same object.

**Weakly-supervised setting.** While these datasets are generated by augmenting existing segmentation ones with descriptive sentences, it could be easier to obtain several referring sentences to the same object than to annotate a dense mask for the objects of interest. In that case we might have a large dataset of the form $\mathcal{D} = \{\mathbf{I}_i, \{\mathbf{T}_{i,j,k}\}_{k=1}^{n_{i,j}^{\mathbf{O}}}\}_{i=1}^{n^{\mathbf{I}}}$, where each object $\mathbf{O}_{i,j}$ of the $n_{ij}^O$ existing objects is implicitly described by the set of referring expressions without *a priori* knowledge of its mask. This is the setup from previous works (Strudel et al., 2022; Liu et al., 2023a; Kim et al., 2023).

**CLIP.** We use the text and image encoders of CLIP (Radford et al., 2021), which we refer to as $\psi_{\mathrm{CLIP}} : \mathcal{T} \to \mathbb{R}^e$ and $\phi_{\mathrm{CLIP}} : \mathbb{R}^{\mathcal{I}} \to \mathbb{R}^e$, respectively.

## 2.2 Related Work

**Fully and Weakly-Supervised Referring Image Segmentation.** The problem of segmenting target objects in images using natural language expressions was first tackled by Hu et al. (2016) using a recurrent neural network. Since then, a variety of *fully-supervised* solutions (i.e., using both referring expressions and pixel-dense masks) have been introduced in the literature (Margffoy-Tuay et al., 2018; Li et al., 2018; Jing et al., 2021; Chen et al., 2022; Ding et al., 2021; Feng et al., 2021; Yang et al., 2022; Liu et al., 2023b). Most recent methods within this category tend to focus on extracting language features using Transformer (Vaswani et al., 2017; Devlin et al., 2018) based models (Jing et al., 2021; Ye et al., 2019; Yang et al., 2022; Ouyang et al., 2023), which are then fused with initial image features obtained using convolutional networks (Jing et al., 2021; Ye et al., 2019) or transformer-based encoders (Yang et al., 2022; Ouyang et al., 2023; Liu et al., 2023b) in a cross-modal pipeline. Wang et al. (2022b) proposed a contrastive pre-training framework similar to CLIP (Radford et al., 2021) to learn separate image and text transformers in a fully-supervised setting. Strudel et al. (2022) proposed TSEG, the first *weakly-supervised* approach to this problem by training a model without the use of pixel-dense segmentation masks, a setting more recently explored by Liu et al. (2023a) and Kim et al. (2023).

**Zero-shot Pixel-Dense Tasks and Referring Image Segmentation.** Recent zero-shot approaches use large-scale pre-pretraining to transfer the learned skills to previously unseen tasks without explicit training. CLIP (Radford et al., 2021) is an example of such an approach on image-text pairs. This idea has also been applied to language-driven pixel-dense prediction tasks, such as open-vocabulary object detection (Gu et al., 2021; Bangalath et al., 2022; Liu et al., 2023c; Minderer et al., 2023) or semantic segmentation (Mukhoti et al., 2023; Chen et al., 2023; Liang et al., 2023); and to class-agnostic instance segmentation (Wang et al., 2022a; 2023; Kirillov et al., 2023). In Yu et al. (2023) the authors introduce the first zero-shot approach to referring image segmentation by combining FreeSolo (Wang et al., 2022a) to generate object instance masks and selecting one using a CLIP-based approach.

## 3 Three-Stage Framework for Referring Image Segmentation

Our approach consists of three stages, as shown in Figure 1. Stages 1 and 2 leverage existing pre-trained models in a zero-shot manner to obtain two sets of masks from the original, mask-less dataset ($\mathcal{D}$ in Figure 1): one containing *all instance masks* of the referred object in the original dataset expressions ($\mathcal{S}_1$ in Figure 1), and the other containing a zero-shot *choice of which mask is referenced* in the expression ($\mathcal{S}_2$ in Figure 1). Both of these are used as input to Stage 3, where we first use set $\mathcal{S}_2$ to pre-train a grounded model, and then use set $\mathcal{S}_1$ (containing all instance masks) within a constrained greedy matching training framework to *bootstrap and correct* zero-shot selection mistakes. Stages 1, 2 and 3 are described in detail in Sections 3.1, 3.2 and 3.3, respectively.

### 3.1 *Segment*: Open-Vocabulary Segmentation from Referring Expressions

The goal of this section is to establish a method to extract all instance segmentation masks for image $\mathbf{I}_i$ given a set of referring expressions $\mathbf{T}_{i,j,k}$ from the $\mathcal{D}$ dataset. Throughout this process, we assume these sentences explicitly include the object being referred in the expression. To achieve this, we introduce a three-step, zero-shot process (presented in Figure 2):

1. **Noun Extraction (NE)**: in a similar fashion to Yu et al. (2023), we use a text dependency parser such as spaCy (Honnibal and Johnson, 2015) to extract the key noun phrase in each of the referring expressions $\mathbf{T}_{i,j,k}$.

2. **Dataset Class Projection (CP)**: using CLIP's text encoder (Radford et al., 2021), we project the extracted noun phrase to a set of objects specific to the dataset context by picking the label which has the maximum similarity with each extracted phrase. For performance reasons, we consider a contextualized version of both the dataset labels and noun phrases, using `"a picture of [CLS]"` as input to $\psi_{\text{CLIP}}$ ahead of computing the embedding similarity.

3. **Open-Vocabulary Instance Segmentation (OS)**: all the projected nouns corresponding to $\mathbf{T}_{i,j,k}, \forall k$ and the image $\mathbf{I}_i$ are then passed to an open-vocabulary instance segmentation model. We obtain it by combining an open-set object detector (e.g., Grounding DINO Liu et al. (2023c)) and a class-agnostic object segmentation model (e.g., SAM (Kirillov et al.,

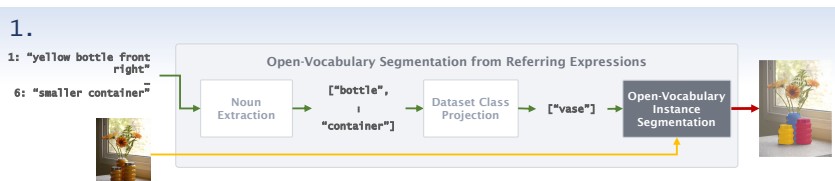

Figure 2: **Open-Vocabulary Segmentation from Referring Expressions**: given a referring expression, we first extract the key noun phrase, project it to a set of context-specific classes, and then use open-vocabulary instance segmentation to obtain all the candidate masks for the object.

2023) or FreeSOLO Wang et al. (2022a)) to obtain all the possible instance segmentation masks for the referring object. The output of this process is a set of pseudo-ground-truth instance segmentation masks for each $\mathbf{T}_{i,j,k}$ defined as $\{\mathbf{m}_{i,j,k}^c\}_1^C$ with $\mathbf{m}_{i,j,k}^c \in [0,1]^{\mathcal{I}}$.

As a result of these steps, we will have successfully constructed set $\mathcal{S}_1$ which will be used in Stages 2 and 3. In Section 4.2 we perform an ablation over each of these steps.

### 3.2 *Select*: ZERO-SHOT CHOICE FOR REFERRING IMAGE SEGMENTATION

Given the mechanism in Stage 1, for each image $\mathbf{I}_i$ and referring expression $\mathbf{T}_{i,j,k}$ we now have a set of binary masks $\{\mathbf{m}_{i,j,k}^1, \ldots, \mathbf{m}_{i,j,k}^C\}$. The goal in Stage 2 is to determine which of these candidate masks corresponds to the single object referred in $\mathbf{T}_{i,j,k}$. Shtedritski et al. (2023) and Yang et al. (2023) observe that CLIP, potentially due to its large training dataset, contains some visual-textual referring information. Yang et al. (2023) note that CLIP when visually prompted using a reverse blurring mechanism (*i.e.*, when everything but the object instance is blurred) achieves good zero-shot performance on the similar task of Referring Expression Comprehension.

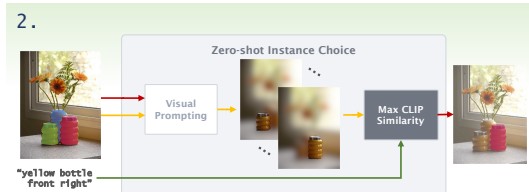

Figure 3: **Zero-Shot Choice for Referring Image Segmentation**: following the main idea from (Yang et al., 2023), we choose a zero-shot mask from the candidate ones by performing a visual prompting to obtain images with the object highlighted via reverse blurring, and then use CLIP similarity to determine the most likely mask choice.

With this insight, we apply the same visual prompting technique to the instance selection problem, as shown visually in Figure 3. Given $\mathbf{T}_{i,j,k}$, we compute its CLIP text embedding, $\psi_{\text{CLIP}}(\mathbf{T}_{i,j,k})$ and choose the mask that satisfies:

$$\max_{c \in \{1,\ldots,C\}} \text{SIM}\left(\phi_{\text{CLIP}}(\mathbf{I}_{i,j,k}^c), \psi_{\text{CLIP}}(\mathbf{T}_{i,j,k})\right), \tag{1}$$

where SIM is the cosine similarity, defined as $\text{SIM}(u,v) = u^\top v/\|u\|\|v\|$ for vectors $u$ and $v$, and $\mathbf{I}_{i,j,k}^c$ is the visually prompted version of $\mathbf{I}_i$ for $\mathbf{m}_{i,j,k}^c$ using the reverse blurring mechanism (with $\sigma = 50$) as in Yang et al. (2023). This effectively constructs the pre-training set $\mathcal{S}_2$ which is used in Stage 3. In Section 4.3 we perform ablations over this stage's zero-shot instance choice pipeline.

### 3.3 *Correct*: CONSTRAINED GREEDY MATCHING FOR WEAKLY-SUPVERVISED RIS

In practice, we have a complete, zero-shot referring image segmentation method using just Stages 1 and 2 of the pipeline. While this might yield good performance already, the zero-shot choice mechanism from Stage 2 will inevitably make mistakes due to a lack of explicit modeling of reference information in the CLIP embedding similarity. We introduce in Stage 3 (Figure 4) a training scheme that attempts to (i) *pre-train* a grounded model with the information already available in the zero-shot chosen masks of Stage 2, and (ii) *correct* some of those mistakes through a constrained greedy matching loss with all the possible masks of Stage 1.

To achieve (i), we simply take set $\mathcal{S}_2$ and train a bootstrapped model (e.g., with the LAVT architecture (Yang et al., 2022)) using a cross-entropy loss. The idea is that the resulting model – referred throughout as ZSBOOTSTRAP – should generalize over the training data, *grounding* it on the concept of referring instructions from the zero-shot outputs of Stage 2.

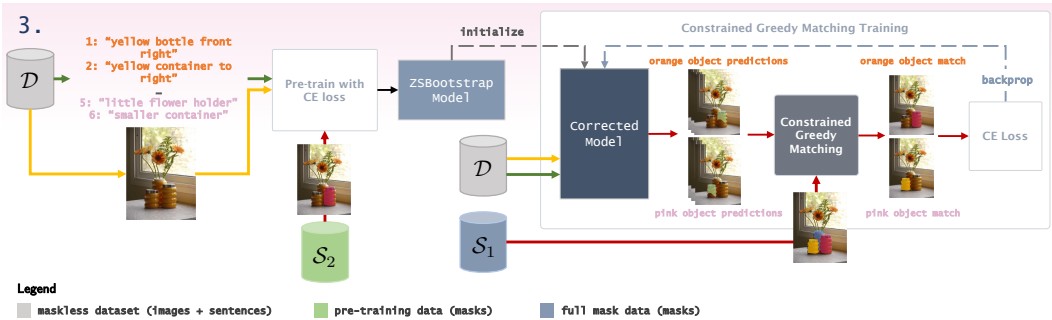

Figure 4: **Grounding + Constrained Greedy Matching**: using set $\mathcal{S}_2$ masks, we start by pre-training a zero-shot bootstrapped model (ZSBOOTSTRAP) that grounds referring concepts which is used to initialize a corrected model trained using set $\mathcal{S}_1$ masks with constrained greedy matching.

To achieve (ii) and correct the mistakes of the zero-shot choice we can use the data from set $\mathcal{S}_1$, which (ideally) contains the instance masks of all the objects of the same category as the referenced one in a scene. Given we do not have access to the ground-truth masks, we cannot know which masks are incorrect. However, from the weakly-supervised dataset format described in Section 2.1, we know that two references corresponding to the same object $\mathbf{O}_{i,j}$ *should have the same mask*, and those corresponding to different objects $\mathbf{O}_{i,j}$ and $\mathbf{O}_{i,j'}$, $j \neq j'$, *should not have the same mask*.

For ease of notation we will drop the image index $i$ for the rest of this section (given the loss is defined per image) and consider $\hat{\mathbf{m}}_{j,k} = f(\mathbf{I}, \mathbf{T}_{j,k})$. We design a loss that simultaneously drives the model towards the most likely mask from set $\mathcal{S}_1$ (*i.e.*, from the set $\{\mathbf{m}_{j,k}^1, \ldots, \mathbf{m}_{j,k}^C\}$), while ensuring that different objects in the same image choose different masks. We take inspiration from the work of Romera-Paredes and Torr (2016), in which the authors use the Hungarian method to solve the bipartite matching problem of outputting segmentation masks using a recurrent network. Similarly, we define our matching problem as:

$$\max_{\delta \in \Delta} \quad \ell_{\text{match}}(\hat{\mathbf{m}}_{j,k}, \{\mathbf{m}_{j,k}^1, \ldots, \mathbf{m}_{j,k}^C\}, \delta) = \sum_{j,k,c} \ell_{\text{IoU}}(\hat{\mathbf{m}}_{j,k}, \mathbf{m}_{j,k}^c)\delta_{j,k,c}, \tag{2}$$

where $\ell_{\text{IoU}}$ is a differentiable IoU loss as defined by Romera-Paredes and Torr (2016), and $\delta_{j,k,c}$ is a binary variable defining whether the mask $\mathbf{m}_{j,k}^c$ for $c \in \{1, \ldots, C\}$ has been matched to object $\mathbf{O}_j$ for all $k$, subject to the constraint that $\delta \in \Delta$ where:

$$\Delta = \left\{ \delta_{j,k,c} \in \{0,1\}, \underbrace{\sum_c \delta_{j,k,c} = 1 \; \forall j, k}_{\text{\textcircled{1} choose one mask per output prediction}}, \underbrace{\delta_{j,k,c} = \delta_{j,k',c} \; \forall c, j, k \neq k'}_{\text{\textcircled{2} every reference of the same object has the same mask}}, \underbrace{\sum_j \delta_{j,k,c} \leq 1, \forall k, c}_{\text{\textcircled{3} references to different objects have different masks}} \right\}.$$

Note that while the perfect matching problem from Romera-Paredes and Torr (2016) admits an optimal solution under Hungarian matching, this is not the case in our setup as the number of set $\mathcal{S}_1$ masks might be smaller than the number of objects if Stage 1 fails to segment one or more instances of the object in the scene. So instead we perform *constrained greedy matching* by taking $j^1, k^1, c^1 = \max l_{\text{IoU}}(\hat{\mathbf{m}}_{j,k}, \mathbf{m}_{j,k}^c)$ across all sentences and candidate masks, and assigning $\delta^*_{j^1,k,c^1} = 1$, for all $k$ – thus guaranteeing \textcircled{2}. $(j^1, c^1)$ then get added to an exclusion set $\mathcal{C}$, and the next matching occurs by considering $j^2, k^2, c^2 = \max_{(j,c) \notin \mathcal{C}} l_{\text{IoU}}(\hat{\mathbf{m}}_{j,k}, \mathbf{m}_{j,k}^c)$ – guaranteeing \textcircled{1} and \textcircled{3}. This process continues until the full matching, $\delta^*$, has been determined (pseudocode presented in Appendix A). While constrained greedy matching does not guarantee the optimality of the problem solution, empirically it often yields the same one as Hungarian matching at a fraction of the running time. With the matching determined, the loss per image becomes:

$$\mathcal{L}_i = \sum_{j,k,c} \mathcal{L}_{\text{CE}}\left(\hat{\mathbf{m}}_{j,k}, \mathbf{m}_{j,k}^c\right)\delta^*_{j,k,c}, \tag{3}$$

where $\mathcal{L}_{\text{CE}}$ is the cross-entropy loss. Assuming $f$ already has some referring information from pre-training, this is expected to improve the performance of the overall model as it will force it to satisfy conditions \textcircled{1}, \textcircled{2} and \textcircled{3}.

Table 1: **Zero-shot and Weakly-Supervised Comparison**: oIoU (top) and mIoU (bottom) results on benchmark datasets for our corrected model trained using constrained greedy matching (S+S+C), as well as our zero-shot (S+S) method, along with the existing baselines GL CLIP (Yu et al., 2023), TSEG (Strudel et al., 2022), TRIS (Liu et al., 2023a), Shatter&Gather (Kim et al., 2023), and LAVT (Yang et al., 2022), and ablations. The first column refers to the type of method: zero-shot (ZS), weakly-supervised (WS) or fully-supervised (FS). Best zero-shot results are highlighted in **purple**, and the best weakly-supervised ones in **green**. For RefCOCOg, U and G refer to the UMD and Google partitions, respectively.

| | | RefCOCO | | | RefCOCO+ | | | RefCOCOg | | |
|---|---|---|---|---|---|---|---|---|---|---|
| | | val | testA | testB | val | testA | testB | val(U) | test(U) | val(G) |
| **oIoU** | | | | | | | | | | |
| ZS | GL CLIP | 24.88 | 23.61 | 24.66 | 26.16 | 24.90 | 25.83 | 31.11 | 30.96 | 30.69 |
| | GL CLIP (SAM) | 24.50 | 26.00 | 21.00 | 26.88 | 29.95 | 22.14 | 28.92 | 30.41 | 28.92 |
| | S+S (*Ours*) | 33.31 | 40.35 | 26.14 | 34.84 | 43.16 | 28.22 | 35.71 | 42.10 | 41.70 |
| WS | TRIS | 31.17 | 32.43 | 29.56 | 30.90 | 30.42 | 30.80 | 36.00 | 36.19 | 36.23 |
| | S+S+C (*Ours*) | 50.13 | 60.70 | 43.46 | 40.61 | 49.68 | 29.54 | 41.96 | 42.59 | 42.18 |
| FS | LAVT | 72.73 | 75.82 | 68.79 | 62.14 | 63.38 | 55.10 | 61.24 | 62.09 | 60.50 |
| **mIoU** | | | | | | | | | | |
| ZS | GL CLIP | 26.20 | 24.94 | 26.56 | 27.80 | 25.64 | 27.84 | 33.52 | 33.67 | 33.61 |
| | GL CLIP (SAM) | 30.79 | 33.08 | 27.51 | 32.99 | 37.17 | 29.47 | 39.45 | 40.85 | 40.66 |
| | S+S (*Ours*) | 36.95 | 43.77 | 27.97 | 37.68 | 46.24 | 29.31 | 41.41 | 47.18 | 47.57 |
| WS | TSEG | 25.95 | – | – | 22.62 | – | – | 23.41 | – | – |
| | Shatter&Gather | 34.76 | 34.58 | 35.01 | 28.48 | 28.60 | 27.98 | – | – | 28.87 |
| | S+S+C (*Ours*) | 56.03 | 64.73 | 38.64 | 46.89 | 55.45 | 33.88 | 48.18 | 48.61 | 49.41 |
| FS | LAVT | 74.46 | 76.89 | 70.94 | 65.81 | 70.97 | 59.23 | 63.34 | 63.62 | 63.66 |

## 4 EXPERIMENTS

The aim of this section is to showcase the effectiveness of our method in closing the gap of zero-shot and weakly-supervised methods with the fully-supervised state-of-the-art using only images and referring sentences. To achieve this, we report results on:

- **Segment+Select** (S+S – *zero-shot*): using the open-vocabulary instance segmentation paired with the zero-shot instance choice to perform zero-shot RIS on each sample of the validation and test datasets (Stages 1 and 2 of Figure 1), and

- **Segment+Select+Correct** (S+S+C – *weakly-supervised*): the full pipeline described in Section 3, including the grounding/pre-training step using set $\mathcal{S}_2$ masks to generate a Zero-shot Bootstrapped (ZSBOOTSTRAP) model and the constrained greedy training stage using set $\mathcal{S}_1$ masks to obtain the final corrected model (Stages 1, 2 and 3 of Figure 1).

To justify the design choices taken at each step, we perform ablations on the open-vocabulary instance mask generation from Stage 1 in Section 4.2, on the zero-shot instance choice mechanism from Stage 2 in Section 4.3, and on the constrained greedy matching loss used in Stage 3 in Section 4.4.

**Datasets and Evaluation.** Following the established literature, we report results on RefCOCO (Yu et al., 2016), RefCOCO+ (Mao et al., 2016) and RefCOCOg (Nagaraja et al., 2016), which have 19,994, 19,992 and 26,711 images in total with 142,210, 141,564, 104,560 referring expressions, respectively. As mentioned in Section 2.1, each image in these datasets includes a certain number of object instances, which in turn include 3 referring expressions each. In Appendix B we analyze the number of object instances per image in the datasets. In terms of evaluation metrics, following previous works we report mean and overall Intersection over Union (mIoU and oIoU, respectively).

**Baselines.** We compare with previously established baselines: the zero-shot method Global-Local CLIP (GL CLIP, (Yu et al., 2023)), the weakly-supervised baselines TSEG (Strudel et al., 2022), TRIS (Liu et al., 2023a) and Shatter&Gather (Kim et al., 2023), and the fully-supervised method LAVT (Yang et al., 2022). The first step of GL CLIP generates a pool of class-agnostic segmentation masks

using FreeSOLO (Wang et al., 2022a), which is followed by a selection step using a global-local CLIP similarity mechanism. Since our method uses SAM (Kirillov et al., 2023) in Stage 1, for fairness of comparison, we also report results on an ablation of GL CLIP which uses SAM to generate the candidate masks – we refer to it as GL CLIP (SAM).

**Implementation Details.** The models trained in Stage 3 follow the cross-modal architecture of LAVT (Yang et al., 2022). They use a BERT (Wolf et al., 2020) encoder, and we initialize the Transformer layers with the weights of the Swin Transformer (Liu et al., 2021) pre-trained on ImageNet-22k (Deng et al., 2009). The optimizer (AdamW), learning rate ($5 \times 10^{-4}$), weight decay ($10^{-2}$) and other initialization details follow the ones of LAVT (Yang et al., 2022), with the exception of the batch size which we set at 60 instead of the original 32. Given we use 4 NVIDIA A40 GPUs (48GB VRAM each), this batch size change leads to significant speed-ups without noticeable performance changes. We pre-train our bootstrapped model (ZSBOOTSTRAP) for 40 epochs, and subsequently train the corrected constrained greedy matching one for a further 40 epochs.

## 4.1 ZERO-SHOT (S+S) AND WEAKLY-SUPERVISED (S+S+C) EXPERIMENTS

In Table 1 we report the performance of our zero-shot method, S+S, our main trained method, S+S+C, and other existing methods and ablations. Overall, our zero-shot and weakly-supervised method outperform the baselines in a majority of the test sets, establishing new state-of-the-art results in both.

**Zero-shot Performance.** We observe that our zero-shot method outperforms the FreeSOLO version of Global-Local CLIP in most of the validation and test sets considered, with oIoU improvements ranging from $1.5\%$ to over $19\%$, and mostly still significantly outperforms GL CLIP (SAM) with improvements as high as $14\%$ in RefCOCO. This highlights that a better class-agnostic instance segmentation method *is not the main driver* behind our improved performance. In several scenarios, we also observe that S+S improves upon the weakly-supervised TRIS and Shatter&Gather.

**Weakly-supervised Performance.** By pre-training and correcting the potential zero-shot mistakes using the constrained greedy loss, the performance improves significantly. Our weakly-supervised method leads to oIoU improvements ranging from $1.3\%$ to over $20\%$ over S+S, all without using any supervised referring segmentation masks. As expected, this is still below the fully-supervised results, yet note that, for example, in RefCOCO+ testA our model's oIoU only lags the fully-supervised LAVT by less than $14\%$, a significant improvement from the previous $33\%$ gap of TRIS in that same test set. In all cases except for oIoU in RefCOCO+ testB, S+S+C establishes a new state-of-the-art in weakly-supervised RIS, outperforming both TRIS and Shatter&Gather by margins as high as $26\%$.

**Efficiency Comparison.** A key difference between the zero-shot baselines (including our S+S) and our full method S+S+C is that the latter requires a "pre-processing" step (Stages 1 and 2) to be applied to the full training set followed by two training steps (Stage 3). This induces a one-time overhead of training our weakly-supervised model which is approximately $2.5\times$ that of the fully-supervised LAVT (with the same architecture). However, once either model is trained, the average inference time per sample is $2.4 - 14\times$ faster than the zero-shot methods and comparable to that of TRIS. A forward pass on a single NVIDIA A40 GPU only takes $0.2s$ compared to $0.49s$, $2.95s$ and $1.78s$ for GL CLIP, GL CLIP (SAM), and S+S+C, respectively. GL CLIP (SAM)

Table 2: **Open-Vocabulary Instance Mask Generation**: ablation of the steps of mask generation Noun Extraction (**NE**), Dataset Class Projection (**CP**) and Class-agnostic Instance Segmentation (**CS**) on the first $1,000$ samples of the RefCOCO training set. Evaluation metrics computed by selecting the highest mIoU mask compared to the ground-truth.

| NE | CP | CS | oIoU | mIoU |
|----|----|----|------|------|
| spaCy | ✓ | SAM | **60.32** | **61.53** |
| nltk | ✓ | SAM | 52.56 | 58.27 |
| spaCy | ✗ | SAM | 48.82 | 43.90 |
| spaCy | ✓ | FreeSOLO | 56.19 | 58.63 |

is $\sim 6\times$ slower than GL CLIP due to the increased inference time and number of candidate masks per dataset sample for SAM vs. FreeSOLO (111 vs. 49 on average, respectively).

## 4.2 ABLATION ON STAGE 1: OPEN-VOCABULARY INSTANCE MASK GENERATION

In this section, we validate our design choices in generating the grounding instance masks using the procedure from Section 3.1. To do so, we perform experiments on RefCOCO by varying the noun extraction mechanism – using nltk (Loper and Bird, 2002) instead of spaCy (Honnibal and Johnson,

2015) –, with or without context dataset projection and ablations on the open-vocabulary instance segmentation module – using FreeSOLO (Wang et al., 2022a) instead of SAM (Kirillov et al., 2023). In Table 2, we present the results of these ablations of Stage 1 of our method evaluated on the first $1,000$ examples of the RefCOCO training set by choosing the closest mask to the ground-truth ones per referring sentence. We observe that our choice performs the best in terms of oIoU and mIoU on this set, and that context dataset projections are an important factor in achieving that performance. Note that while Yu et al. (2023) put the FreeSOLO upper bound limit on RefCOCO val set at $42.08\%$ oIoU, we achieve significantly better masks with FreeSOLO ($56.19\%$ oIoU) since we do not query the method on the full image, but rather a cropped version around the bounding box initially obtained by GroundingDINO (part of step 3 described in Section 3.1).

## 4.3   ABLATION ON STAGE 2: ZERO-SHOT INSTANCE CHOICE

To study the effect of the zero-shot instance choice in producing good grounding masks for our model given the masks produced by Stage 1 from Section 3.1, we perform ablations using different visual prompting mechanisms – Red Ellipse from (Shtedritski et al., 2023) and Reverse Blur from (Yang et al., 2023) – with different CLIP visual backbones, and compare them to simple baselines like randomly choosing the right mask (Random) or by accessing the ground-truth ones (Oracle). The results are presented in Table 3. Note that on average there are 3.65 object instances per image, so as expected Random achieves approximately $1/3$ of the performance of Oracle. With the exception of Oracle, which is inaccessible at inference time, the Reverse Blur approach from (Yang et al., 2023) with a CLIP ViT-L/14@336px visual backbone

Table 3: **Zero-shot Instance Choice**: ablation of the zero-shot instance choice options on the first $1,000$ examples of the RefCOCO training set. Oracle is in gray as it provides a benchmark by comparing to the inaccessible at inference time ground-truth masks (copied from Table 2). ViT-B32 and ViT-L14 refer to the ViT-B/32 and ViT-L/14@336px CLIP visual backbones.

| Choice | Prompt | Backbone | oIoU | mIoU |
|--------|--------|----------|------|------|
| Oracle | – | – | 60.32 | 61.53 |
| Random | – | – | 23.92 | 24.32 |
| ZS | Red Ellipse | ViT-B32 | 24.84 | 26.34 |
|    |             | ViT-L14 | 32.91 | 34.77 |
| ZS | Reverse Blur | ViT-B32 | 34.35 | 35.97 |
|    |              | ViT-L14 | **36.88** | **38.46** |

outperforms all other approaches, validating our choice at the level of Stage 2. However, the gap between the best and our zero-shot choice highlights there is room for improvement in Stage 3.

A natural question at this point is whether we can correct the zero-shot choice earlier than the training time at Stage 3, and then simply train a model on those masks. To test this, we apply the constrained greedy matching algorithm by replacing $\ell_{\text{match}}$ from Eq. 2 with the CLIP similarity from Eq. 1 to the generation of masks for the first $1,000$ training set examples of RefCOCO, obtaining an oIoU of 36.34 and an mIoU of 38.11. This suggests that attempting to correct the zero-shot choice is not nearly as effective as our Stage 3.

## 4.4   ABLATION ON STAGE 3: CORRECTION VIA CONSTRAINED GREEDY MATCHING

With the goal of understanding how effective constrained greedy matching is to the performance of our method, we ablate over that second training step of Stage 3. Starting from the pre-trained model using set $\mathcal{S}_2$ masks, ZSBOOTSTRAP, we report in Table 4 the effect of training on those masks for 40 further epochs to match the total compute used in our method (+40 EPOCHS) as well as training with constrained greedy matching (S+S+C)

Table 4: **Constrained Greedy Matching Ablation**: comparison of fine-tuning ZSBOOTSTRAP (pre-trained on set $\mathcal{S}_2$ masks) using the same zero-shot selected masks for 40 further epochs (+40 EPOCHS) and constrained greedy matching (S+S+C) on RefCOCO.

| | oIoU | | | mIoU | | |
| | val | testA | testB | val | testA | testB |
|---|---|---|---|---|---|---|
| ZSBOOTSTRAP | 33.61 | 42.20 | 26.12 | 37.29 | 44.18 | 28.43 |
| +40 EPOCHS | 34.62 | 44.47 | 25.87 | 38.62 | 46.34 | 28.44 |
| S+S+C | **50.13** | **60.70** | **43.46** | **56.03** | **64.73** | **38.64** |

performance on RefCOCO. Observe that training for 40 epochs on set $\mathcal{S}_2$ masks leads to a minimal increment over the baseline performance, while allowing the model to choose the greedy match from all the instance masks (set $\mathcal{S}_1$ masks) via the loss from Section 3.3 leads to a boost in performance.

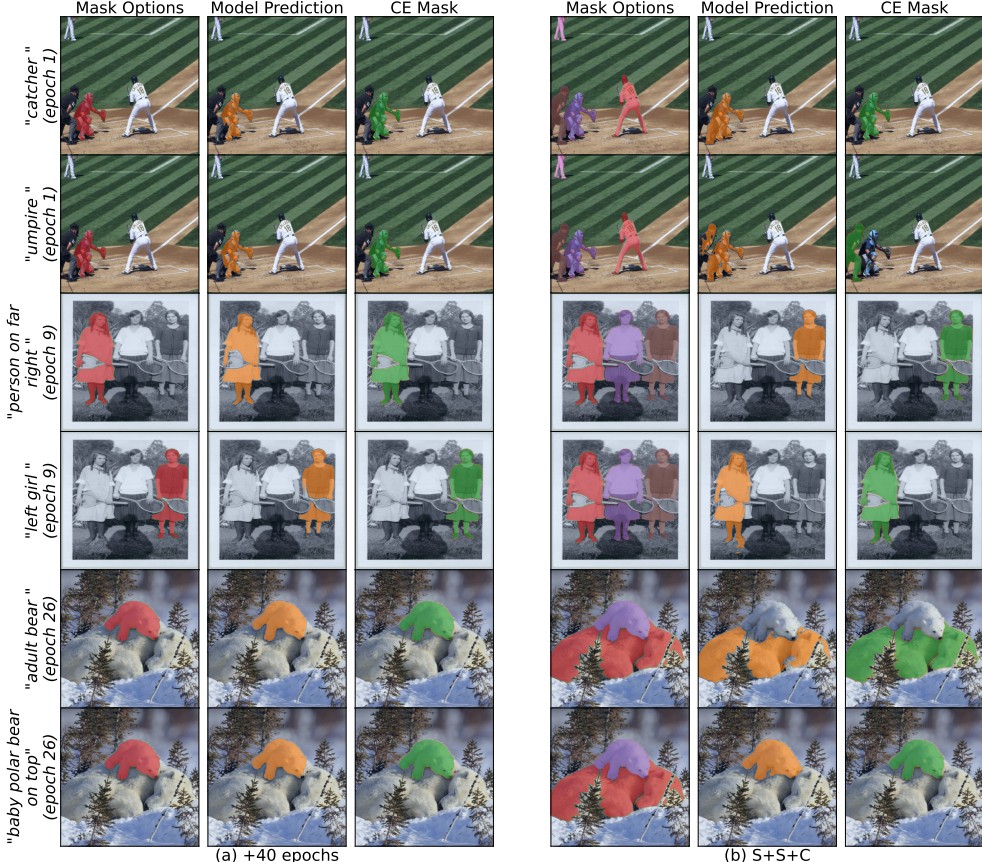

Figure 5: **Qualitative Constrained Greedy Matching Ablation**: qualitative training set examples of the RefCOCO dataset with all the masks available (Mask Options), the model's output (Model Prediction) and the mask which will be used in the cross-entropy loss term in the training (CE Mask). In (a) +40 EPOCHS, the training is limited to the single zero-choice mask, which in this case is incorrect for one example per pair. Our constrained greedy matching loss in (b) (S+S+C) can choose between different instances of the class, allowing it to correct the initial zero-shot error.

To qualitatively understand the significant improvement brought by constrained greedy matching, in Figure 5 we showcase RefCOCO training dataset examples where S+S was originally wrong for one of them. For example, in rows 1 and 2 of Figure 5 both *"catcher"* and *"umpire"* are matched to the catcher by S+S, which means the +40 EPOCHS model is always forced to choose that mask (CE Mask is always that zero-shot one). By contrast, our constrained greedy matching loss allows the model to choose from all the players in the field, such that when the catcher mask is matched to the prediction from row 1, *"umpire"* is now matched to the umpire mask given the mIoU is greater with that one than with any of the remaining ones. By allowing the model to choose between the training masks in greedy matching, the model is able to recover from the incorrect zero-shot choice in these cases – this effect is compounded over the training epochs as better matching initially leads to quicker correction of future mistakes.

## 5 CONCLUSIVE REMARKS

We propose a three stage pipeline for weakly-supervised RIS that obtains all the instance masks for the referred object (*segment*), gets a good first guess on the right one using a zero-shot instance choice method (*select*), and then bootstrap and corrects it through the constrained greedy matching loss (*correct*). Our zero-shot method (S+S) outperforms other zero-shot baselines by as much as $19\%$, and our full method (S+S+C) sets the new state-of-the-art for this task, reducing the gap between weakly-supervised and fully-supervised models to as little as $13\%$ in some cases from nearly $33\%$ with TRIS (Liu et al., 2023a). Despite the need for a one-off cost for training, inference on our weakly-supervised model is $2.4 - 14\times$ faster than other competitive zero-shot baselines.

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

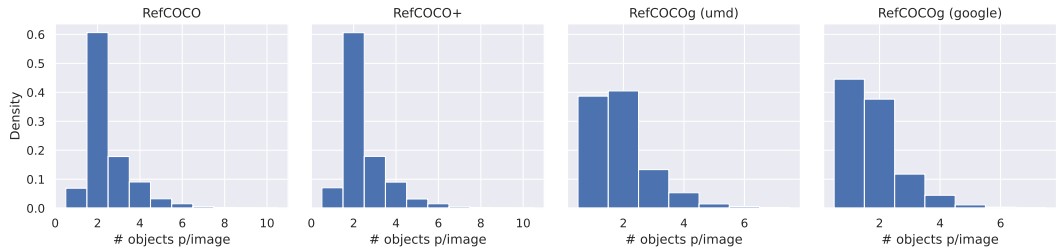

Figure 6: **Object Instances per Image**: histograms of the distribution of the number of object instances referenced in each image within the training sets of the studied datasets.

## A  CONSTRAINED GREEDY MATCHING ALGORITHM

The details of our constrained greedy matching algorithm are presented in 1. For the sake of simplicity of notation, the image index $i$ is dropped as this matching only occurs for all objects within an image.

---

**Algorithm 1** Constrained Greedy Matching

---

1: **Input:** mask choices $\{\mathbf{m}_{j,k}^c\}_c$, model predictions $\hat{\mathbf{m}}_{j,k}$
2: **Result:** greedy matching $\delta_{j,k,c}^*$
3: $\delta_{j,k,c}^* \leftarrow 0$
4: $\mathcal{C} \leftarrow \emptyset$                              ▷ Initialize the exclusion set
5: $\mathcal{M} \leftarrow \left\{ j,k,c : \ell_{\mathrm{IoU}}\left( \hat{\mathbf{m}}_{j,k}, \mathbf{m}_{j,k}^c \right) \right\}$     ▷ Compute pseudo-IoU for all mask & prediction pairs
6: $\mathrm{SORT}(\mathcal{M})$                         ▷ Sort them in descending order
7: **while** $\mathcal{M} \neq \emptyset$ **do**
8:     $j', k', c' \leftarrow \mathrm{POP}(\mathcal{M})$              ▷ Get the next highest IoU mask choice
9:     **if** $c \in \mathcal{C}$ or $j \in \mathcal{C}$ **then**     ▷ If either the object or mask has been matched, skip it
10:         **continue**
11:     **end if**
12:     **for** $k$ **do**                         ▷ Match it for all expressions of the same object
13:         $\delta_{j',k,c'}^* \leftarrow 1$
14:     **end for**
15:     $\mathcal{C} \leftarrow \mathcal{C} \cup \{(j', c')\}$     ▷ Add object & mask to the exclusion set to avoid re-matching them
16: **end while**
17: **return** $\delta_{j,k,c}^*$

---

## B  NUMBER OF OBJECT INSTANCES PER IMAGE

In Figure 6 we show the distribution of the number of objects per image in the training sets of RefCOCO, RefCOCO+ and RefCOCOg (umd and google splits). Note that this average is higher for RefCOCO and RefCOCO+ than for RefCOCOg. The higher the average number of object instances per image, the more effective we expect the loss from Stage 3 to be in correcting the zero-shot mistakes due to the matching mechanism. This intuition is backed by the experimental results presented in Section 4.1 of the main paper, where the improvement in RefCOCO and RefCOCO+ from S+S+C is higher than the one in RefCOCOg (umd and google partitions).

## C  PIXEL-LEVEL CONTRASTIVE ABLATION

On top of the constrained greedy matching results, we also experimented with a pixel-dense contrastive loss term. This contrastive term, $\mathcal{L}_{\leftrightarrow}$ has the goal of regularizing the output by explicitly leveraging the insight that references to the same object (positive examples) in an image should have the same output, and that other objects in the same image (negative examples) should have a different output.

Table 5: **Zero-shot and Weakly-Supervised Comparison with $\mathcal{L}_\leftrightarrow$**: oIoU (top) and mIoU (bottom) results on benchmark datasets for our zero-shot method (S+S), our full method (S+S+C), the zero-shot bootstrapped models from Stage 3 (ZSBOOTSTRAP), and an ablation with the proposed pixel-dense contrastive loss $\mathcal{L}_\leftrightarrow$, along with existing baselines and ablations. The first column refers to the type of method: zero-shot (ZS), weakly-supervised (WS) or fully-supervised (FS). Best zero-shot results are highlighted in **purple**, and the best weakly-supervised ones in **green**. For RefCOCOg, U refers to the UMD partion, and G refers to the Google partion.

| | | RefCOCO | | | RefCOCO+ | | | RefCOCOg | | |
| --- | --- | --- | --- | --- | --- | --- | --- | --- | --- | --- |
| | | val | testA | testB | val | testA | testB | val(U) | test(U) | val(G) |
| **oIoU** | | | | | | | | | | |
| | GL CLIP | 24.88 | 23.61 | 24.66 | 26.16 | 24.90 | 25.83 | 31.11 | 30.96 | 30.69 |
| ZS | GL CLIP (SAM) | 24.50 | 26.00 | 21.00 | 26.88 | 29.95 | 22.14 | 28.92 | 30.41 | 28.92 |
| | S+S (*Ours*) | 33.31 | 40.35 | 26.14 | 34.84 | 43.16 | 28.22 | 35.71 | 42.10 | 41.70 |
| | TRIS | 31.17 | 32.43 | 29.56 | 30.90 | 30.42 | 30.80 | 36.00 | 36.19 | 36.23 |
| WS | ZSBOOTSTRAP (*Ours*) | 33.61 | 42.20 | 26.12 | 34.13 | 42.03 | 26.60 | 38.27 | 40.09 | 37.03 |
| | S+S+C (*Ours*) | 50.13 | 60.70 | 43.46 | 40.61 | 49.68 | 29.54 | 41.96 | 42.59 | 42.18 |
| | S+S+C + $\mathcal{L}_\leftrightarrow$ (*Ours*) | 50.43 | 61.66 | 43.28 | 39.47 | 48.97 | 30.08 | 41.62 | 42.48 | 42.26 |
| FS | LAVT | 72.73 | 75.82 | 68.79 | 62.14 | 63.38 | 55.10 | 61.24 | 62.09 | 60.50 |
| **mIoU** | | | | | | | | | | |
| | GL CLIP | 26.20 | 24.94 | 26.56 | 27.80 | 25.64 | 27.84 | 33.52 | 33.67 | 33.61 |
| ZS | GL CLIP (SAM) | 30.79 | 33.08 | 27.51 | 32.99 | 37.17 | 29.47 | 39.45 | 40.85 | 40.66 |
| | S+S (*Ours*) | 36.95 | 43.77 | 27.97 | 37.68 | 46.24 | 29.31 | 41.41 | 47.18 | 47.57 |
| | TSEG | 25.95 | – | – | 22.26 | – | – | 23.41 | – | – |
| | Shatter&Gather | 34.76 | 34.58 | 35.01 | 28.48 | 28.60 | 27.98 | – | – | 28.87 |
| WS | ZSBOOTSTRAP (*Ours*) | 37.29 | 44.18 | 28.43 | 38.84 | 46.13 | 29.60 | 43.41 | 44.78 | 42.72 |
| | S+S+C (*Ours*) | 56.03 | 64.73 | 38.64 | 46.89 | 55.45 | 33.88 | 48.18 | 48.61 | 49.41 |
| | S+S+C + $\mathcal{L}_\leftrightarrow$ (*Ours*) | 55.46 | 64.45 | 38.54 | 46.56 | 55.96 | 34.61 | 48.53 | 48.71 | 49.84 |
| FS | LAVT | 74.46 | 76.89 | 70.94 | 65.81 | 70.97 | 59.23 | 63.34 | 63.62 | 63.66 |

However, note that for negative examples considered pairwise, the output of the network should only be distinct in locations where the instance chosen masks from Equation 3 are active, as it should be 0 elsewhere for both. Given the full matching $\delta^*$, we can obtain the chosen mask for each input as $\mathbf{m}_{i,j,k} = \sum_c \mathbf{m}^c_{i,j,k}\delta^*_{j,k,c}$. Now consider a pair of negative examples for an object $j, k$ and $j^-, k^-$. The "active" pixels in either of the chosen, pseudo-ground-truth masks as $\mathbf{A}(i, j, k, j^-, k^-) = \mathbf{m}_{i,j,k} \cup \mathbf{m}_{i,j^-,k^-}$. These are then used to slice each of the masks such that $\tilde{\mathbf{m}}_{i,j,k} = \hat{\mathbf{m}}_{i,j,k}|_{A(i,j,k,j^-,k^-)}$ (and similarly for $j^-, k^-$). We can then write the contrastive loss as:

$$\mathcal{L}_\leftrightarrow(i, j, k) = \sum_{k^+ \neq k} \mathrm{KL}(\hat{\mathbf{m}}_{i,j,k} \| \hat{\mathbf{m}}_{i,j,k+}) + \sum_{j^- \neq j, k^-} \left[\gamma \mathrm{KL}(\tilde{\mathbf{m}}_{i,j,k} \| \tilde{\mathbf{m}}_{i,j-,k-})\right]^{-1}, \qquad (4)$$

where $\gamma \in \mathbb{R}^+$ is a hyperparameter to tune the balance of negative and positive examples in the contrastive term.

We present the full results in Table 5. As can be observed, the $\mathcal{L}_\leftrightarrow$ loss does not lead to an clear improvement of the results, which is why we have decided to omit it from the main method.

