# OpenReview forum: "Segment, Select, Correct: A Framework for Weakly-Supervised Referring Segmentation"
_ICLR.cc/2024/Conference — Submitted to ICLR 2024_

### Official Review · Reviewer_VXus · 2023-10-31

**Soundness:** 2 fair
**Presentation:** 1 poor
**Contribution:** 2 fair
**Rating:** 5
**Confidence:** 3

**Summary:**

This paper proposed a straightforward pipeline for referring image segmentation through only language sentences. The pipeline is designed to be trained with only images and sentences without masks. To this end, the authors proposed a pipeline of three stages, that is segment, select, and correct, with existing foundation models, such as CLIP, SAM, etc. Experimental results somewhat demonstrate the effectiveness of the proposed method.

**Strengths:**

The setting is interesting. The proposed pipeline is straightforward and in some perspective shows the power of the combination of existing foundation models.

**Weaknesses:**

This paper is a bit difficult to read. It abusively uses colorful dots to represent almost everything, including results, modules, and stages, which heavily hinders reading. This proposed pipeline is a straightforward combination of existing foundation models, what's the insight beyond the combination? In addition, some important key details about the correction stage are not clear, which requires further explanation.

**Questions:**

1. How do you get m^hat and m^c, is the m^hat output of LAVT? In addition, as the authors mentioned in sec3.3, the ZSBootstrap uses LAVT architecture, why the LAVT trained with texts and their pseudo visual masks work better and serve as an error correction model?

2. What does "grounding" mean exactly? What's the difference between referring instance segmentation and grounding? I found the authors use them two both in the paper.

3. Since LAVT is already a referring segmentation method, the authors re-trained it. Does that mean stage 1 and 2 are not valuable once the zsbootstrap model is trained?

---

> ### Author Response · Authors · 2023-11-17
> **Authors' Response to Reviewer VXus**
>
> We thank the reviewer for the time spent on the paper as well as for their comments.
>
> The main idea behind our zero-shot approach is the decomposition of RIS into the segmentation of all objects belonging to the desired target (*segment* in Stage 1) followed by a selection step to choose the right mask to return (*select* in Stage 2). This simple idea realized by combining pre-trained models is shown to be highly effective in practice, beating other published zero-shot baselines by as much as 19% in some evaluation sets.
>
> By leveraging the richness of all the segmentation masks from the *segment* stage, we are able to *correct* a weakly-supervised model trained on the outputs of our zero-shot approach and obtain new state-of-the-art performance in this setting, reducing the gap between fully and weakly-supervised methods from 33% (in previously published work) to as little as 14% in some cases.
>
> **On presentation and notation.** Our goal by introducing the dots and colors to represent datasets and stages in our framework was to simplify the process of reading the paper. However, we understand how this could be overwhelming from the perspective of the readers, so in the updated manuscript we have replaced the original dataset (gray) by $\mathcal{D}$ (and defined it in Section 2.1), the full masks dataset (blue) by $\mathcal{S}_1$ to identify it is the output of Stage 1, and the selected masks dataset (green) by $\mathcal{S}_2$ to identify it is the output of Stage 2. These are defined in the beginning of Section 3. We hope these changes improve the readability of the paper.
>
> **On the contributions beyond combination.** First, while it is true that stage 1 and 2 of our method are mostly combinations of pre-trained methods, these have not been proposed together in this way to solve the problem of Referring Image Segmentation in a zero-shot setting. This approach, while simple, achieves state-of-the-art in the benchmarked datasets, beating more complicated strategies like GL-CLIP from Yu et al., (2023) by as much as 19% in some cases. We believe this is one of our contributions.
>
> Secondly, the main contribution of the paper is the design of a new objective function and training algorithm (constrained greedy matching) in Stage 3. As recognized by Reviewer g3AQ, this is a novel contribution from our paper, yielding significant gains and establishing new state-of-the-art results in the weakly-supervised setting.
>
> **On the unclear key details of the correction stage.** The reviewer mentions that there are ‘some important key details about the correction stage that are not clear, which requires further explanation.’ We would appreciate it if the reviewer could clarify exactly what is unclear about this section.
>
> If the clearness issue comes from the definition of index $c$ as per the question from Reviewer g3AQ, we clarify that $c$ is the index of each binary mask $m^{c}\_{j,k}$ in the set of masks for each reference in the image, i.e., it is an index for the set $\\{m^1_{j,k}, \dots, m^C_{j,k}\\}$ where $C$ is the total number of masks. To avoid confusion we will use the extended set notation throughout the paper, and refer to $c \in \\{1,\dots,C\\}$. We have clarified this at the end of Section 3.1 and at the beginning of Sections 3.2 and 3.3 in the revised manuscript. If there is anything else that is unclear please let us know.

---

> > ### Author Response · Authors · 2023-11-17
> > **Authors' Response to Reviewer VXus (continued)**
> >
> > **(Q1) $\hat{m}$ and the LAVT architecture for the RIS models.** $\hat{m}$ is defined in the 4th paragraph of Section 3.3 as the output of the S+S+C model we are training, which follows a LAVT architecture in our experiments. ZSBootstrap, which is trained on references + pseudo-masks selected in Stage 2, works as a good initialization for the model training in Stage 3 using constrained greedy training.
> >
> > **(Q2) On the use of “grounding”.** To solve referring image segmentation, a method has to understand language and references with respect to images. As such, there is a small high-level difference between RIS and grounding, in that RIS requires language grounding at the model level. Our usage of grounding follows the high-level definition and usage from previous literature (Yu et al., 2023).
> >
> > **(Q3) The importance of Stage 1 and Stage 2.** The goal of the full method is to output a model S+S+C trained on a weakly-supervised dataset that can perform referring image segmentation. Stage 3 of our framework requires the outputs of both Stage 1 and 2 for training, with the output of Stage 1 needed to *correct* the ZSBootstrapped model and obtain S+S+C (our final model). Once this model is obtained, inference is done with a simple forward pass in this model, so Stage 1 and 2 do not need to be run for each inference – as highlighted in the ‘Efficiency Comparison’ paragraph of Section 4.1, this is more efficient than running those stages.
> >
> > We believe we have answered the reviewers’ questions and clarified the weaknesses. We hope this is useful to the reviewer in reconsidering the score attributed, and look forward to continuing the discussion.

---

> > > ### Author Response · Authors · 2023-11-21
> > > **Additional comments or questions**
> > >
> > > Given the rebuttal process is coming to an end in a couple of days, we hope the reviewer has found our answers to be informative and useful to the reviewing process. If there are any outstanding comments or questions, we are happy to clarify them at this point. If we have addressed the issues raised, we would kindly request the reviewer to update their score.

---

> > > > ### Comment · Reviewer_VXus · 2023-11-22
> > > >
> > > > I appreciate the efforts of the authors. The rebuttal clears some of my confusions.  I still suggest to carefully use “grounding” considering that it is not the same task with referring segmentation. And I suggest the authors should carefully design the framework for better illustration.  Considering a major revision for above points, I would like to keep my previous decision.

---

### Official Review · Reviewer_g3AQ · 2023-10-31

**Soundness:** 3 good
**Presentation:** 2 fair
**Contribution:** 2 fair
**Rating:** 5
**Confidence:** 3

**Summary:**

This paper proposes a framework for weakly supervised referring segmentation. It includes 3 steps, open-vocabulary segmentation, selection based on the CLIP model, and correction with a constrained greedy matching mechanism. The intuition behind the idea is interesting. Very strong performance is also obtained.

**Strengths:**

+ The use of CLIP for zero-shot matching is sensible.

+ The overall learning process in step 3 is interesting. It is fully exploiting the dataset information: referring expressions for the same object should lead to the same mask, while  referring expressions for different objects should lead to different mask.

+ The obtained performance is strong.

**Weaknesses:**

-  Notations are not well defined, making it really difficult to understand the details some sections. For example,  in {m^c_{i,j,k} }^c, it is really confusing what 'c' means here.

-  In general, I can appreciate the general idea of the learning mechanism for step 3.  It is really difficult to understand the equation (2) and (3). The reason could be unclear definitions, such as 'c'. Probably there are other undefined notations.

- Fig.1 fails to provide the basic intuition of the third step, which, in my opinion, is the most valuable part of this paper.

**Questions:**

- In the ABLATION part, some experiences are based on the training set of the dataset (Table 2 & Table 3). What is the performance on the val and test sets?

- Stage 3 is designed to correct the mistakes of the zero-shot choice with weak information. Why not use this information directly in stage 2 but take an extra large grounding model from LAVT?

---

> ### Author Response · Authors · 2023-11-17
> **Authors' Response to Reviewer g3AQ**
>
> We thank the reviewer for the time spent on the paper as well as for their comments.
>
> **(W1 & W2) On notation.** We thank the reviewer for bringing these details to our attention. $c$ is the index of each binary mask $m^{c}\_{j,k}$ in the set of masks for each reference in the image, i.e., it is an index for the set $\\{m^1_{j,k}, \dots, m^C_{j,k}\\}$ where $C$ is the total number of masks. To avoid confusion we will use the extended set notation throughout the paper, and refer to $c \in \\{1,\dots,C\\}$.
>
> This change should also clarify the notation from equations 2 and 3. Equation 2 corresponds to the optimization problem we are trying to solve, where the goal is to choose the assignment $\delta_{j,k,c} \in \\{0, 1\\}$ (i.e., either that as mask is chosen or not) for each of the possible ground-truth masks in $\\{m^1_{j,k}, \dots, m^C_{j,k}\\}$ such that the sum of the intersections over unions of the *chosen* masks is maximized (with the matching conditions described by the constraint $\Delta$ defined below Equation 2). Once this problem is solved through greedy input matching to obtain $\delta^*_{j,k,c}$, the actual loss we use is defined in Equation 3, penalizing the cross-entropy of the matched ground-truth mask and the network output $\hat{m}_{j,k}$.
>
> We have clarified this at the end of Section 3.1 and at the beginning of Sections 3.2 and 3.3 in the revised manuscript. If there is anything else that is still unclear in those equations please let us know. Thank you once again for pointing this out.
>
> **(W3) On the intuition of Stage 3 from Figure 1.** We agree with the reviewer that this step is crucial to the success of our framework. The idea of Figure 1 is to give an overview of the whole pipeline where, for conciseness, we avoid providing excessive details on each of the steps. To highlight the intuition and details of Stage 3, we use Figure 4, which we believe gives readers a better understanding of the process in detail. For the final version of the paper, we will attempt to introduce more details of this stage in Figure 1 directly without overloading it.
>
> **(Q1) Ablations on validation and test sets.** The ablation studies in Section 4.2 (Stage 1) and 4.3 (Stage 2) are used as cross-validation to justify the design choices for those stages and, as such, including validation and test set results could be seen as overfitting to those datasets. Instead, we make the decisions on those stages using a small subset of 1,000 samples from the training set and evaluate them against the fully-supervised ground-truth available on that dataset alone, before evaluating the full method on the validation and test sets (Section 4.1).
>
> **(Q2) On correcting in Stage 2.** This is a good point, and one that we explore in the last paragraph of Section 4.3 (copied here):
> > “A natural question at this point is whether we can correct the zero-shot choice earlier than the training time at Stage 3, and then simply train a model on those masks. To test this, we apply the constrained greedy matching algorithm by replacing $\ell_{\text{match}}$ from Eq. 2 with the CLIP similarity from Eq. 1 to the generation of masks for the first 1, 000 training set examples of RefCOCO, obtaining an oIoU of 36.34 and an mIoU of 38.11. **This suggests that attempting to correct the zero-shot choice is not nearly as effective as our Stage 3**.”
>
> On top of this insight we also note that inference on our final S+S+C model takes only $0.2s$ compared to $1.78s$ per sample, so introducing a LAVT-based model is also beneficial in terms of inference running time.
>
> We believe we have answered the reviewers’ questions and clarified the weaknesses. We hope this is useful to the reviewer in reconsidering the score attributed, and look forward to continuing the discussion.

---

> > ### Author Response · Authors · 2023-11-21
> > **Additional comments or questions**
> >
> > Given the rebuttal process is coming to an end in a couple of days, we hope the reviewer has found our answers to be informative and useful to the reviewing process. If there are any outstanding comments or questions, we are happy to clarify them at this point. If we have addressed the issues raised, we would kindly request the reviewer to update their score.

---

### Official Review · Reviewer_xVs3 · 2023-11-01

**Soundness:** 2 fair
**Presentation:** 2 fair
**Contribution:** 2 fair
**Rating:** 3
**Confidence:** 5

**Summary:**

This paper presents a weakly supervised framework for the task of referring image segmentation.
It contains three major components:  a segment component obtaining instance masks for the object mentioned in the referencing instruction, a select component using zero-shot learning to select the correct mask for the given instruction, and a correct component bootstrapping a model to fix the mistakes of zero-shot selection.
Further experiments show good performance compared to previous methods.

**Strengths:**

+ The paper is easy to follow.
+ The performance is good in zero-shot and weakly supervised setup.

**Weaknesses:**

- The paper is a good engineering work, but lacks of enough novelty.
- The experiment is incomplete and not convincing.
- It lacks enough ablation studies on the effectiveness of each components used.
- It lacks enough details on the pre-training bootstrapped model.
- It lacks enough reference and comparison to previous methods.

**Questions:**

What is the major contribution of the proposed framework, since all components are borrowed from existing work?

---

> ### Author Response · Authors · 2023-11-17
> **Authors' Response to Reviewer xVs3**
>
> Respectfully, we believe this review falls well below the high review expectations from a conference such as ICLR. **None of the weaknesses stated is specific enough to be addressed.** We would request further details from the reviewer to be able to rebut the points, as the comments provided are generic and unhelpful in improving the paper.

---

### Official Review · Reviewer_6s9c · 2023-11-03

**Soundness:** 2 fair
**Presentation:** 3 good
**Contribution:** 1 poor
**Rating:** 3
**Confidence:** 4

**Summary:**

This paper proposes a weakly supervised method for referring image segmentation. Starting from class-agnostic object proposals, the proposed method first extracts masks corresponding to the object class which is referenced by the given text, using the open-vocabulary segmentation technique. Then, select the actually referenced mask using the existing zero-shot prompting method. Finally, to refine the obtained mask, the authors propose a correction method by using the assumption that they know if the two references indicate the same object or not. The proposed method obtains better performance than the existing methods.

**Strengths:**

+ This paper addresses important problem that can effectively reduces the annotation cost for referring image segmentation.

+ The paper is overall well written.

**Weaknesses:**

- My major concern is that the proposed method is significantly overfitted to specific datasets.

-- First, the correction method in Section 3.3 requires a significantly strong assumption. The authors assume that multiple references tend to indicate a single object (mask), and they know which references actually correspond to the same mask. However, this assumption will not work for another dataset, and especially, knowing whether two references point to the same object or not is infeasible for a weakly supervise setting.

-- If I understood correctly, dataset class projection in Section 3.1 is to determine which class among the 80 classes in COCO corresponds to the key noun. The COCO class list is carefully curated by human, and each of the 80 classes is mutually exclusive. In the real open-world setting, assuming the specific set of classes is infeasible.

- My second concern is the novelty. Methods for obtaining object proposals, and matching those proposals with zero-shot prompting, are already well-explored techniques for the same research field.

- More baselines should be included.

-- GroundingDINO already conducted referring object detection. GroundingDION + SAM can be directly used for referring image segmentation.

-- For a fairer comparison with Yu et al, FreeSOLO+Select method without Grounding DINO.

-- SAM Proposals (w/o Grounding DINO) + ReverseBlur prompting

**Questions:**

- Why testB performance is signifcantly lower than that of testA?

- Excepting the correction method, please include the ablation studies on zero-shot validation and test sets for all datasets.

---

> ### Author Response · Authors · 2023-11-17
> **Authors' Response to Reviewer 6s9c**
>
> We thank the reviewer for the time spent on the paper as well as for their comments.
>
> The main idea behind our zero-shot approach is the decomposition of RIS into the segmentation of all objects belonging to the desired target (*segment* in Stage 1) followed by a selection step to choose the right mask to return (*select* in Stage 2). This simple idea realized by combining pre-trained models is shown to be highly effective in practice, beating other published zero-shot baselines by as much as 19% in some evaluation sets.
>
> By leveraging the richness of all the segmentation masks from the *segment* stage, we are able to *correct* a weakly-supervised model trained on the outputs of our zero-shot approach and obtain new state-of-the-art performance in this setting, reducing the gap between fully and weakly-supervised methods from 33% (in previously published work) to as little as 14% in some cases.
>
> **(W1) On multiple references of the same object.** We do not believe the assumption of different references for objects in the same image is a strong one. While in the studied datasets the multiple references were the result of multiple human annotations, it is easy to imagine that a dataset can be automatically generated to have multiple references to the same object, given some of these references could even be synthetically generated using a causal language model (e.g., following Brown et al., (2020) or Touvron et al., (2023)). Given the quality of some of the references present in the commonly used for evaluation (RefCOCO, RefCOCO+ and RefCOCOg), it is likely that such a system might yield plausible references that are potentially better than human annotators. The key takeaway is that this assumption, while small, yields strong improvements at the level of performance as shown by our experiments.
>
> **(W1) On the dataset class projection.** We start by noting we do not claim to tackle an open-world setting in this paper, as highlighted by the evaluation section which trains and tests on specific, widely used datasets. While testing these systems in an open-world setting is an interesting open question, it is outside the scope of our paper and method so we leave it for future work.
>
> Our goal is to improve performance in the absence of ground-truth referred segmentation masks. Within this setting – which is a common one for many other applications – having the knowledge of a broad, mutually exclusive set of classes of interest is not a major barrier. It is an assumption often made in many applications such as autonomous driving. Note also that this set of classes can be more extensive than the references in the dataset, as is the case with RefCOCO/RefCOCO+/RefCOCOg which only reference object instances that are a subset of the full COCO class set.
>
> **(W2) Novelty.** First, while it is true that stage 1 and 2 of our method are mostly combinations of pre-trained methods, these have not been proposed together in this way to solve the problem of Referring Image Segmentation in a zero-shot setting. On the particular point from the reviewer that *“methods for obtaining object proposals, and matching those proposals with zero-shot prompting, are already well-explored techniques”*, we highlight that our method differs from GL-CLIP from Yu et al. (2023) which generates mask proposals for **all objects** in the scene, as opposed to our approach which generates masks for **all objects of the target noun phrase**. As shown in the paper results and in the further baselines the reviewer proposed (see below), this key difference is novel and important for the success of our zero-shot method which achieves state-of-the-art in the task. We believe this is one of our main contributions.
>
> Secondly, the main contribution of the paper is the design of a new objective function and training algorithm (constrained greedy matching) in Stage 3. As recognized by Reviewer g3AQ, this is a novel contribution from our paper, yielding significant gains and establishing new state-of-the-art results in the weakly-supervised setting.

---

> > ### Author Response · Authors · 2023-11-17
> > **Authors' Response to Reviewer 6s9c (continued)**
> >
> > **(W3) Baselines.** The reviewer proposed including 3 further zero-shot baselines to our experiments: GroundingDINO + SAM for RIS (taking the highest scoring SAM mask after choosing the highest scoring GroundingDINO bounding box), FreeSOLO + Select and SAM proposals + ReverseBlur selection. We present below oIoU and mIoU results of those 3 baselines on the RefCOCO validation set:
> >
> > | ZS Method | oIoU (RefCOCO val) | mIoU (RefCOCO val) |
> > | :---- | :----: | :----: |
> > | GL-CLIP | 24.88 | 26.20 |
> > | GL-CLIP (SAM) | 24.50 | 30.79 |
> > | GroundingDINO + SAM | 29.95 | 33.98 |
> > | SAM + Select | 18.54 | 22.39 |
> > | FreeSOLO + Select | 23.18 | 24.30 |
> > | Segment + Select (Ours) | **33.31** | **36.95** |
> >
> > We believe these experiments show the benefits of our approach. For the proposal baselines considered (SAM and FreeSOLO), we also show below the “oracle”-choice performance (i.e., choosing the closest to the ground-truth mask):
> >
> > | ZS Method | oIoU (RefCOCO val) | mIoU (RefCOCO val) |
> > | :---- | :----: | :----: |
> > | SAM + Oracle | 54.36 | 55.91 |
> > | FreeSOLO + Oracle | 36.93 | 40.91 |
> >
> > This highlights that the worse-performance of SAM + Select above is mostly attributed to the selection mechanism rather than the mask generation process – as is the case of GL-CLIP (SAM).
> >
> > Thank you for suggesting these zero-shot baselines as they make the experiment section more thorough. Due to time and resource constraints we could not run all the experiments for the rebuttal, but we will add the baselines evaluated on all datasets to the final version of the paper.
> >
> > **(Q1) testB performance vs. testA.** We believe this might be a bias introduced by GroundingDINO and SAM, which might be significantly more successful at detecting people (in testA) than all other objects (in testB). This behavior is also typically observed in fully supervised such as LAVT.
> >
> > **(Q2) Ablations on validation and test sets.** The ablation studies in Section 4.2 (Stage 1) and 4.3 (Stage 2) are used as cross-validation to justify the design choices for those stages and, as such, including validation and test set results could be seen as overfitting to those datasets. Instead, we make the decisions on those stages using a small subset of 1,000 samples from the training set and evaluate them against the fully-supervised ground-truth available on that dataset alone, before evaluating the full method on the validation and test sets (Section 4.1).
> >
> > However, we will include RefCOCO+ and RefCOCOg ablation results from Stage 3 (Section 4.3) to the final version of the paper (currently running).
> >
> > We believe we have answered the reviewers’ questions and clarified the weaknesses. We hope this is useful to the reviewer in reconsidering the score attributed, and look forward to continuing the discussion.
> >
> > Additional references:
> > - Brown, Tom, et al. "Language models are few-shot learners." Advances in neural information processing systems 33 (2020): 1877-1901.
> > - Touvron, Hugo, et al. "Llama 2: Open foundation and fine-tuned chat models." arXiv preprint arXiv:2307.09288 (2023).

---

> > > ### Author Response · Authors · 2023-11-21
> > > **Additional comments or questions**
> > >
> > > Given the rebuttal process is coming to an end in a couple of days, we hope the reviewer has found our answers to be informative and useful to the reviewing process. If there are any outstanding comments or questions, we are happy to clarify them at this point. If we have addressed the issues raised, we would kindly request the reviewer to update their score.

---

### Meta-Review · Area_Chair_DDme · 2023-12-05

**Metareview:**

After discussion, major concerns about the novelty , in-depth analysis and comprehensive comparison with existing works remain. After carefully reading the paper, the review comments, the AC deemed that the paper should undergo a major revision, thus is not ready for publication in the current form.

**Justification For Why Not Higher Score:**

N/A

**Justification For Why Not Lower Score:**

N/A

---

### Decision · Program_Chairs · 2024-01-16

Reject